# Predicting ventricular tachycardia circuits in patients with arrhythmogenic right ventricular cardiomyopathy using genotype-specific heart digital twins

Yingnan Zhang[1,2], Kelly Zhang[1,2], Adityo Prakosa[1,2], Cynthia James[3], Stefan L Zimmerman[4], Richard Carrick[3], Eric Sung[1,2], Alessio Gasperetti[3], Crystal Tichnell[3], Brittney Murray[3], Hugh Calkins[3], Natalia A Trayanova[1,2]*

[1]Department of Biomedical Engineering, Johns Hopkins University, Baltimore, United States; [2]Alliance for Cardiovascular Diagnostic and Treatment Innovation, Johns Hopkins University, Baltimore, United States; [3]Division of Cardiology, Department of Medicine, Johns Hopkins Hospital, Baltimore, United States; [4]Department of Radiology, Johns Hopkins University, Baltimore, United States

*For correspondence: ntrayanova@jhu.edu

**Abstract** Arrhythmogenic right ventricular cardiomyopathy (ARVC) is a genetic cardiac disease that leads to ventricular tachycardia (VT), a life-threatening heart rhythm disorder. Treating ARVC remains challenging due to the complex underlying arrhythmogenic mechanisms, which involve structural and electrophysiological (EP) remodeling. Here, we developed a novel genotype-specific heart digital twin (Geno-DT) approach to investigate the role of pathophysiological remodeling in sustaining VT reentrant circuits and to predict the VT circuits in ARVC patients of different genotypes. This approach integrates the patient's disease-induced structural remodeling reconstructed from contrast-enhanced magnetic-resonance imaging and genotype-specific cellular EP properties. In our retrospective study of 16 ARVC patients with two genotypes: plakophilin-2 (*PKP2*, $n = 8$) and gene-elusive (GE, $n = 8$), we found that Geno-DT accurately and non-invasively predicted the VT circuit locations for both genotypes (with 100%, 94%, 96% sensitivity, specificity, and accuracy for GE patient group, and 86%, 90%, 89% sensitivity, specificity, and accuracy for *PKP2* patient group), when compared to VT circuit locations identified during clinical EP studies. Moreover, our results revealed that the underlying VT mechanisms differ among ARVC genotypes. We determined that in GE patients, fibrotic remodeling is the primary contributor to VT circuits, while in *PKP2* patients, slowed conduction velocity and altered restitution properties of cardiac tissue, in addition to the structural substrate, are directly responsible for the formation of VT circuits. Our novel Geno-DT approach has the potential to augment therapeutic precision in the clinical setting and lead to more personalized treatment strategies in ARVC.

## eLife assessment

This **important** study brings together a clear application of the digital twin approach to make predictions using patient specific models with different genotypes. The data are **compelling** and go beyond the current state-of-the-art to support proof-of-principle evidence. Given the low subject numbers, further studies will be required going forward to support the veracity of the data and its translational utility.

## Introduction

Arrhythmogenic right ventricular cardiomyopathy (ARVC) is an inherited cardiac disease that affects young adults and has a prevalence estimated as high as 1 in 1000 (*Sen-Chowdhry et al., 2010*). ARVC is a major cause of ventricular tachycardia (VT), a life-threatening fast heart rhythm that can lead to sudden cardiac death (SCD) and accounts for up to 10% of unexplained SCD cases in people younger than 65 years old (*Calkins et al., 2017*; *Thiene et al., 2007*). Catheter ablation, which delivers energy to disrupt abnormal electrical conduction, is a mainstay in treating sustained VT in ARVC, but the success rate of ablation in ARVC patients is only 50–80% (*Mathew et al., 2019*; *Arbelo and Josephson, 2010*) and there is a high rate of VT recurrence (*Waintraub and Gandjbakhch, 2020*). Determining all the locations that could sustain VTs is always challenging in treating ARVC and requires extensive mapping. Difficulty in identifying and targeting all VT circuits could lead to VT recurrence as early as a few months after the initial procedure (*Souissi et al., 2018*; *Mathew et al., 2019*). If all VT circuits could be localized and defined prior to ablation, procedural duration could be potentially shortened, and the effectiveness of the procedure in ARVC patients could be significantly enhanced.

Most ARVC cases are associated with mutations in desmosomal genes, including plakophilin-2 (*PKP2*), desmoglein-2 (*DSG2*), desmocollin-2 (*DSC2*), plakoglobin (*JUP*), and desmoplakin (*DSP*) (*Dalal et al., 2005*). Among these, *PKP2* is the most common genotype, observed in 60% to 78% of ARVC patients with known pathogenic variants (*Mahdieh et al., 2018*; *Protonotarios et al., 2022*; *Corrado et al., 2020*). The PKP2 protein plays a significant role in maintaining the structural integrity of the ventricular myocardium and in facilitating signal transduction pathways, thus *PKP2* pathogenic variants lead to fibrotic remodeling and subsequently to distorted electrical conduction (*Vimalanathan et al., 2018*). Interestingly, around one-third of all ARVC patients do not have known causal pathogenic variants (*Protonotarios et al., 2022*; *Corrado et al., 2020*; *James et al., 2021*). The pathogenesis of these gene-elusive (GE) patients is highly associated with frequent high-intensity exercises that lead to fibrosis formation on the RV (*Benito et al., 2011*; *Breuckmann et al., 2009*; *Sawant et al., 2014*). Overall, both structural and genotype-modulated electrical abnormalities serve as essential VT substrates in ARVC (*El-Battrawy et al., 2018*); however, their specific roles in VT circuit formation remain unexplored.

Previous studies from our group have successfully employed patient-specific heart digital twins in investigating VT mechanisms and in predicting VT locations and morphologies to support ablation targeting in ischemic heart diseases (*Prakosa et al., 2018*; *Deng et al., 2019*; *Sung et al., 2020*; *Sung et al., 2021*). Here, we develop a new personalized genotype-specific heart digital twin approach, Geno-DT, which combines image-based structural information with genotype-specific EP properties at the cell and organ levels. We use the approach to investigate the role of pathophysiological remodeling in ARVC in sustaining VT reentrant circuits and to predict the VT circuits in ARVC patients of the two main genotypes, *PKP2* and GE. The results of this study advance the understanding of arrhythmogenesis in ARVC and offer a pathway to improving VT targeting by ablation.

## Results

The Geno-DT approach involves creating three-dimensional (3D) patient-specific electrophysiological (EP) ventricular models incorporating both personalized structural remodeling (diffuse fibrosis and dense scar) constructed from late gadolinium enhancement cardiac magnetic resonance (LGE-CMR) and genotype-specific (GE and *PKP2*) cellular-level membrane kinetics developed here based on the patient's genetic testing results. For 16 ARVC patients of the two genotypes, *PKP2* and GE, we analyzed VT induction in each personalized heart model to understand the arrhythmogenic mechanisms, and specifically, the contributions of structural remodeling and genotype-modulated EP alterations to sustaining VT reentrant circuits in ARVC. The Geno-DT approach was used to predict rapid-pacing induced VT circuit locations in these patients, which would constitute the ideal targets for ablation. The accuracy of the predictions was demonstrated by comparison to the VT circuits induced in clinical EP studies (EPS). An overview of our study is presented in *Figure 1*.

### Patient characteristics

This retrospective study was approved by the institutional review board and included 16 patients with ARVC. *Table 1* provides demographic information for this cohort. All patients were adults with

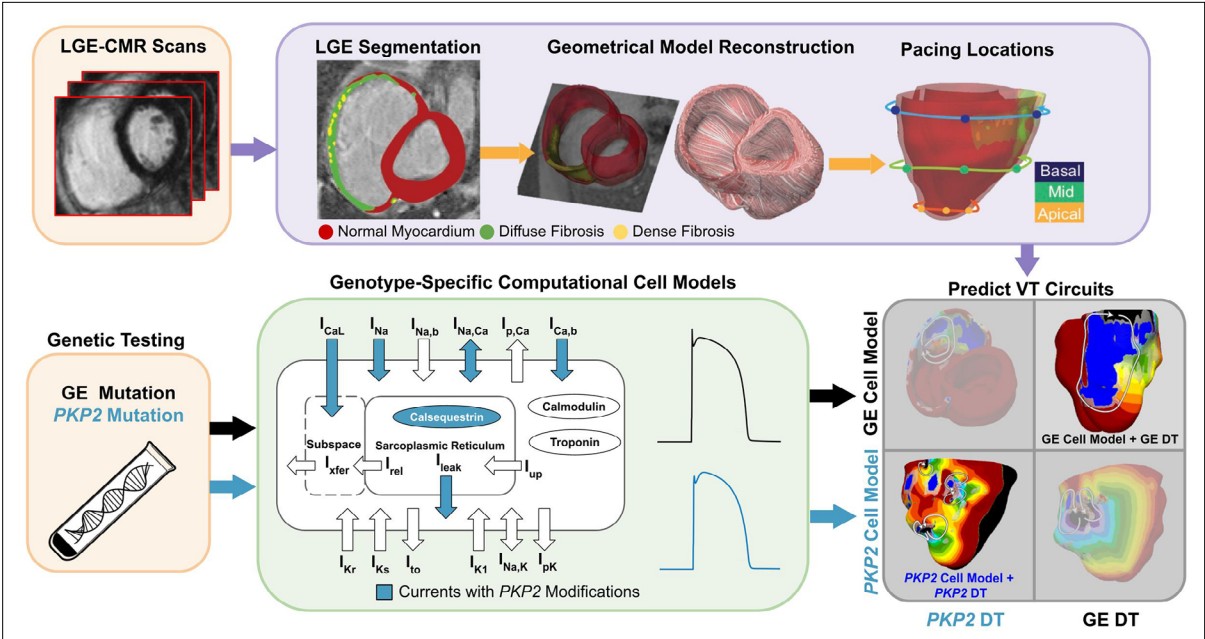

**Figure 1.** Overview of the study. The flowchart summarizes the workflow of using Geno-DT to understand ARVC arrhythmogenesis and to predict VTs in ARVC patients of the two genotype groups, *PKP2* and GE. Geno-DT integrates genotype-specific cell models (green) and patient-specific clinical-image-based heart digital twin modeling. Orange blocks refer to clinical data, which include genetic testing results and LGE-MRI images for each patient in the cohort. Patient-specific geometrical heart models were reconstructed from the LGE-CMR (top, purple, left and middle images). Genotype-specific cell models (green) developed here were incorporated into each heart model based on the patient's genetic testing result. The integrated multi-scale (from the subcellular to the organ), patient-specific Geno-DT models were subjected to rapid pacing from multiple sites on the RV (top purple, rightmost image) to understand the role of remodeling in ARVC arrhythmogenesis and to predict VT circuits (bottom gray).

a median age of 31 years, and 75% of them were female. All had VTs induced during clinical EPS, where information about the patients' VTs were recorded. Genetic testing for ARVC risk variants was performed for the entire cohort. Eight patients were found to have *PKP2* loss-of-function variants, while the remaining eight did not test positive for variants in any of the known causal genes and were thus considered gene-elusive (GE; *James et al., 2021*).

All 16 patients had RV enhancement on LGE-CMR; none of them exhibited enhancement on the left ventricle (LV). The clinical parameters previously found to be associated with VT in ARVC (RV ejection fraction and RV end-diastolic volume index [RVEDVI]) (*te Riele et al., 2014*) did not differ significantly between the GE and *PKP2* groups. There was also no statistically significant difference in any of the other common clinical features between the two genotype groups.

## EP properties of the *PKP2* cell model

Our Geno-DT approach incorporates, in the patient-specific computational models, cellular EP that is distinct between the two genotypes, *PKP2* and GE. The cell-level electrical behavior for the GE genotype was modeled based on the *ten Tusscher and Panfilov, 2006* (TT2) human ventricular model (see Methods). Since there is currently no cell model representation of *PKP2* pathogenic variant that can be used in organ-level simulation in terms of computational tractability, we developed one here, by modifying the GE cell model to represent remodeling in the sodium currents and calcium-handling as reported in experimental studies (*Sato et al., 2009*; *Kim et al., 2019*; *Lyon et al., 2021*). Details on the *PKP2* cell model implementation can be found in Materials and methods.

The new *PKP2* model includes a downscaled maximum conductance for the sodium current ($I_{Na}$), resulting in a peak inward sodium current at a membrane potential of –36 mV that is truncated from –502.71 pA/pF in the GE model to –171.56 pA/pF in the *PKP2* model (*Figure 2A*). Additionally, the *PKP2* cell model includes various calcium currents, each modified from the baseline GE model (see Methods). These changes collectively produce an altered calcium transient (CaT) with a larger area under the curve and a faster decay, giving it a more acute shape (*Figure 2B*). The CaT in the *PKP2* cell

**Table 1.** Patient characteristics ($n = 16$).

| Clinical characteristics | ARVC GE patients ($n = 8$) | ARVC *PKP2* patients ($n = 8$) | p value |
|---|---|---|---|
| Male | 2 {25} | 2 {25} | - |
| Age at CMR, years | 31.0 [22-45] | 35.3 [18-55] | 0.55 |
| Age at first clinical VT, years | 34.0 [22-45] | 31.0 [17-55] | 0.94 |
| ICD implantation | 7 {88} | 7 {88} | - |
| Beta blocker | 8 {100} | 5 {63} | 0.056 |
| Sodium channel blocker | 2 {25} | 3 {38} | 0.59 |
| Syncope | 2 {25} | 2 {25} | - |
| Cardiac function | | | |
| RV hypokinesis | 4 {50} | 6 {75} | 0.15 |
| RVEF (%) | 37.6±5.5 | 36.1±9.7 | 0.76 |
| RVCO (L/min) | 4.7±1.2 | 4.9±0.9 | 0.72 |
| RVEDVI (ml/m²) | 131.8±31.2 | 144.5±63.3 | 0.66 |
| LVEF (%) | 57.2±6.5 | 55.7±10.0 | 0.74 |
| LVEDVI (ml/m²) | 89.3±9.2 | 89.7±23.0 | 0.97 |
| LVCO (L/min) | 5.1±0.8 | 5.0±0.9 | 0.86 |

Values are given as *n* {%}, mean [range], or mean ± standard deviation. p Values were calculated using Student's *t*-test and z-test with p ≤ 0.05 as statistically significant. CMR = cardiac magnetic resonance; VT = ventricular tachycardia; ICD = implantable cardiac defibrillator; RV = right ventricle; LV = left ventricle; RVEF = right ventricular ejection fraction; RVCO = right ventricular cardiac output; RVEDVI = right ventricular end-diastolic volume index; LVEF = left ventricular ejection fraction; LVCO = left ventricular cardiac output; LVEDVI = left ventricular end-diastolic volume index. The cardiac function parameters were obtained from CMR reports.

model has a longer time to peak (32 vs 27 ms), a significantly higher peak (0.0023 vs 0.00084 mM), and a shorter time to 90% return from peak (319 vs 379 ms).

The resulting action potential (AP) shape of the *PKP2* cell is distinct from that of the GE cell, as depicted in *Figure 2C*. The downregulation of $I_{Na}$ results in a lower upstroke peak and slower maximum upstroke velocity ($V_{max}$). The changes to calcium handling led to an elevated resting membrane potential and longer AP duration at 90% depolarization ($APD_{90}$). These differences in AP behavior between the two cells underlie the organ-level EP differences between *PKP2* and GE genotypes.

Since in this study we used a rapid pacing protocol to induce VTs in the personalized ARVC models, cell-level restitution properties, and specifically the difference in restitution properties between the *PKP2* and GE cell models, have important consequences for ARVC arrhythmogenesis. *Figure 3* examines these restitution properties. The GE cell model exhibited a decrease in $APD_{90}$ with diastolic interval (DI) shortening (*Figure 3A*), consistent with normal calcium-handling behavior (*Franz, 2003*). In contrast, the *PKP2* cell model's AP showed some shortening of phase two duration but with very little reduction in plateau amplitude, resulting in more triangular APs at fast pacing without much change in total duration.

*Figure 3B* highlights the differences in restitution curves for the *PKP2* and GE cell models. Notably, at the fast-pacing intervals of the steep initial phase (25–160 ms), the two restitution curves diverge significantly. While the GE cell $APD_{90}$ continued to decrease to 148ms before loss-of-capture, the *PKP2* cell $APD_{90}$ only decreased to 208 ms. The maximum slopes of the restitution curves were measured to be 1.16 and 0.59 for the GE and *PKP2* cell models, respectively. The substantially lower slope for the *PKP2* cell's restitution curve indicated a reduced ability of its AP to adapt to fast pacing rates.

The GE and *PKP2* cell models described above were incorporated in LGE-based geometrical models of the patients' ventricles from the respective patient groups. In addition to EP remodeling, ARVC is also characterized by structural remodeling, i.e the presence of dense scar and diffuse fibrosis. While replacement scar is non-conductive, myocardium in the diffuse fibrosis regions exhibits altered EP properties, which were represented in the personalized ventricular models by additional alterations in

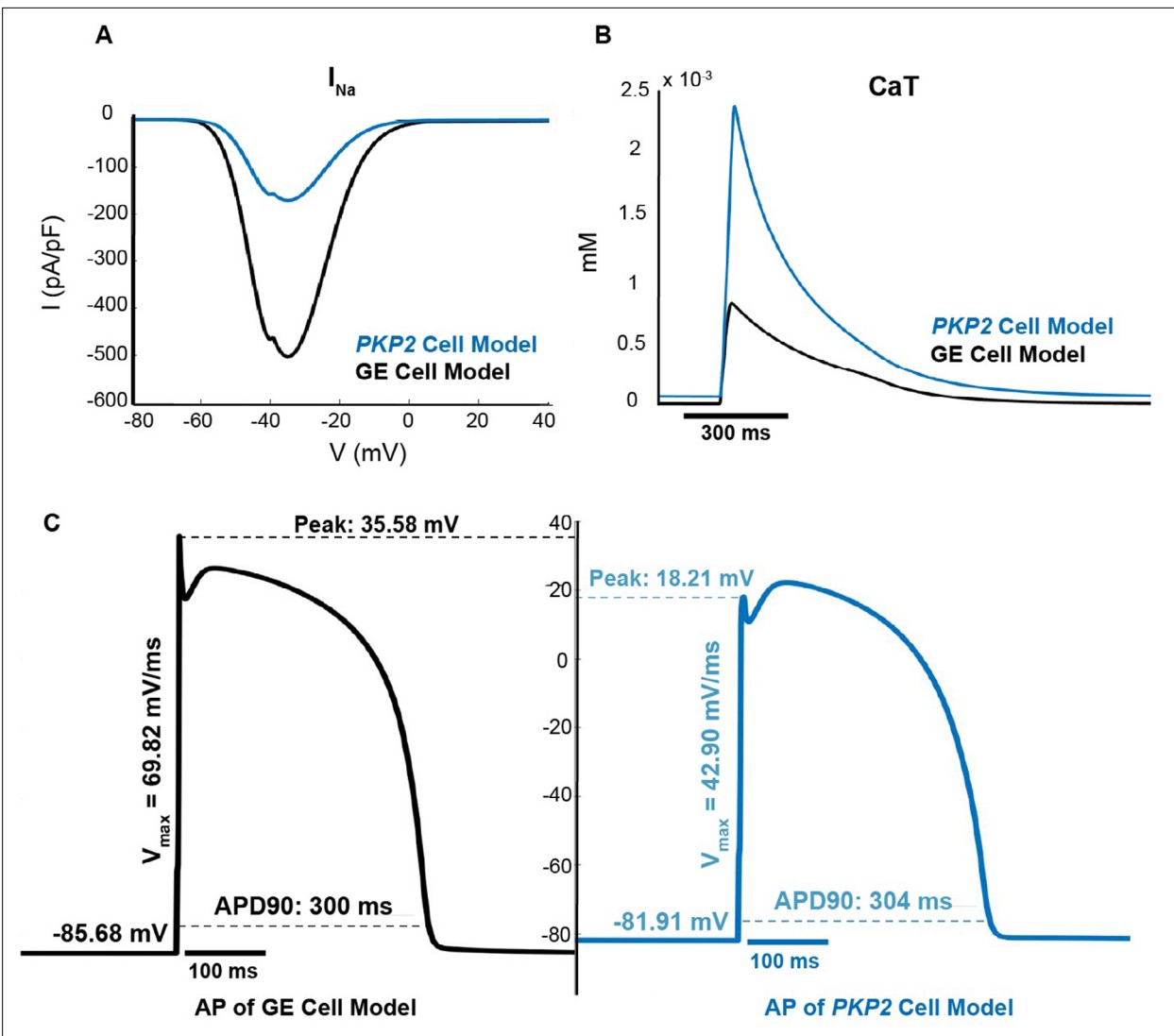

**Figure 2.** The *PKP2* cell model has lower excitability and altered calcium-cycling as compared to the GE model. (**A**) Current-voltage curves for $I_{Na}$; (**B**) Calcium transient curves; (**C**) Action potentials at steady state at 1 Hz pacing.

cell-level properties in these regions, as we have done previously (*Shade et al., 2020*; *O'Hara et al., 2022*); see Materials and methods for detail. The resulting heart models of ARVC patients thus represented both EP and structural remodeling in ARVC.

## Analysis of structural remodeling distribution and its relationship with VT inducibility for different genotype patient groups

Here, we first utilized the LGE-reconstructed ventricular geometrical models to examine the distribution of structural remodeling in various regions of the RV and its relationship to VT. We found that across the entire ARVC cohort, the diffuse fibrosis was most abundant at the base of the RV and least abundant at the RV apex (*Figure 4A*). There were significant differences in the amount of diffuse fibrosis between the basal, mid, and apical regions (14.01 ± 4.37% for basal RV vs 5.41 ± 2.61% for mid RV vs 1.13 ± 1.71% for apical RV, p < 0.0001 for pairwise comparisons). A similar trend was observed for dense scar amounts with significant differences between the basal, mid, and apical regions (5.51 ± 2.71% vs 1.74 ± 1.99% vs 0.42 ± 0.90%, p < 0.05 for pairwise comparisons; *Figure 4B*). There was no significant difference when comparing the amounts of structural remodeling between anterior, lateral, and posterior regions.

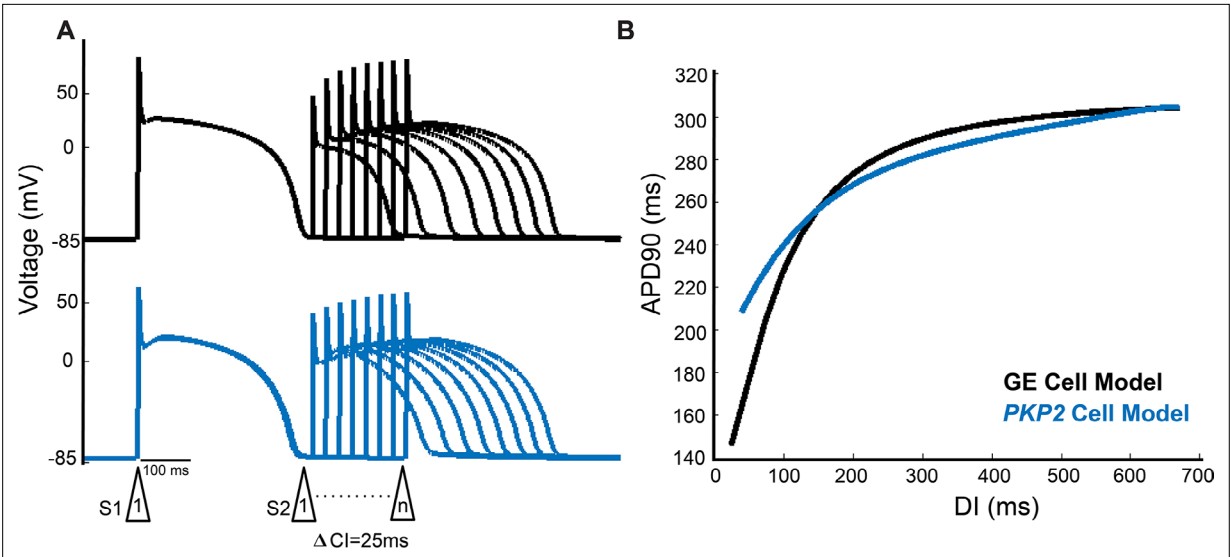

**Figure 3.** The *PKP2* cell model exhibits poorer rate adaptation and a flatter electrical restitution curve as compared to the GE model. (**A**) Representative premature APs at DIs decrementing to just before loss of one-to-one capture of GE cell model (top) and *PKP2* cell model (bottom). (**B**) APD restitution curves of GE and *PKP2* cell models.

We found that the distribution of diffuse fibrosis was different in the two patient groups, with GE patients having a significantly higher amount of diffuse fibrosis burden than *PKP2* patients in the basal anterior wall (6.07 ± 3.18% vs 1.85 ± 0.017%, p < 0.01 for pairwise comparisons) and basal lateral wall (5.77 ± 2.34% vs 3.24 ± 1.40%, p < 0.05 for pairwise comparisons), and a significantly lower amount in the basal posterior wall (3.36 ± 2.92% vs 7.73 ± 4.38%, p < 0.05 for pairwise comparisons; *Figure 4C*). There was no statistical significance in the dense scar distribution between the two genotype groups.

Next, we performed simulations, with the personalized Geno-DT heart models, of VT induction following rapid pacing (see Materials and methods) and evaluated the induced VT circuits in the entire cohort. We performed a total of 144 ventricular simulations (16 patients × 9 pacing locations) to determine the induced VT circuit morphologies. We examined the correlations between the structural remodeling features (diffuse fibrosis volume [DF], dense scar volume [DS], total fibrotic remodeling volume (both scar and diffuse fibrosis) [TFV]) and the number of unique VT morphologies induced during EPS across the different RV AHA segments. *Figure 5* summarizes pairwise correlations between these variables for both GE and *PKP2* patient groups. Our analysis revealed a highly positive correlation between the number of VTs induced during EPS and the volume of each type of structural remodeling (both DF and DS) and total fibrotic volume (TFV) in the GE patient group (r = 0.91, 0.91, 0.92 for DF, DS and TFV respectively), while in the *PKP2* group, we observed only moderate correlation (r = 0.65, 0.7, 0.7 for DF, DS and TFV respectively).

Importantly, we found a strong correlation between VTs induced during EPS and those induced in simulations within the different RV regions in both patient groups (r = 0.99 for both groups), indicating that our heart digital twins effectively capture the occurrence pattern of EPS-induced VTs. Notably, although the structural remodeling in the *PKP2* group was less correlated with EPS VTs as compared to the GE group, the VTs induced in the *PKP2* models still exhibited high correlation with the VTs recorded during EPS.

These findings suggest that structural remodeling plays a more important role in VT arrhythmogenesis in GE patients than in *PKP2* patients. The moderate correlation in the *PKP2* group indicates that additional factors contribute to VT, specifically the EP remodeling associated with the *PKP2* pathogenic variants.

## Predicting VT circuits using Geno-DT

We next assessed the predictive capability of the Geno-DT approach in determining accurately the VT circuit numbers and locations in the two ARVC genotype groups. In *Figure 6*, the bullseye plots summarize the number of unique VT morphologies induced during EPS and those induced in Geno-DTs

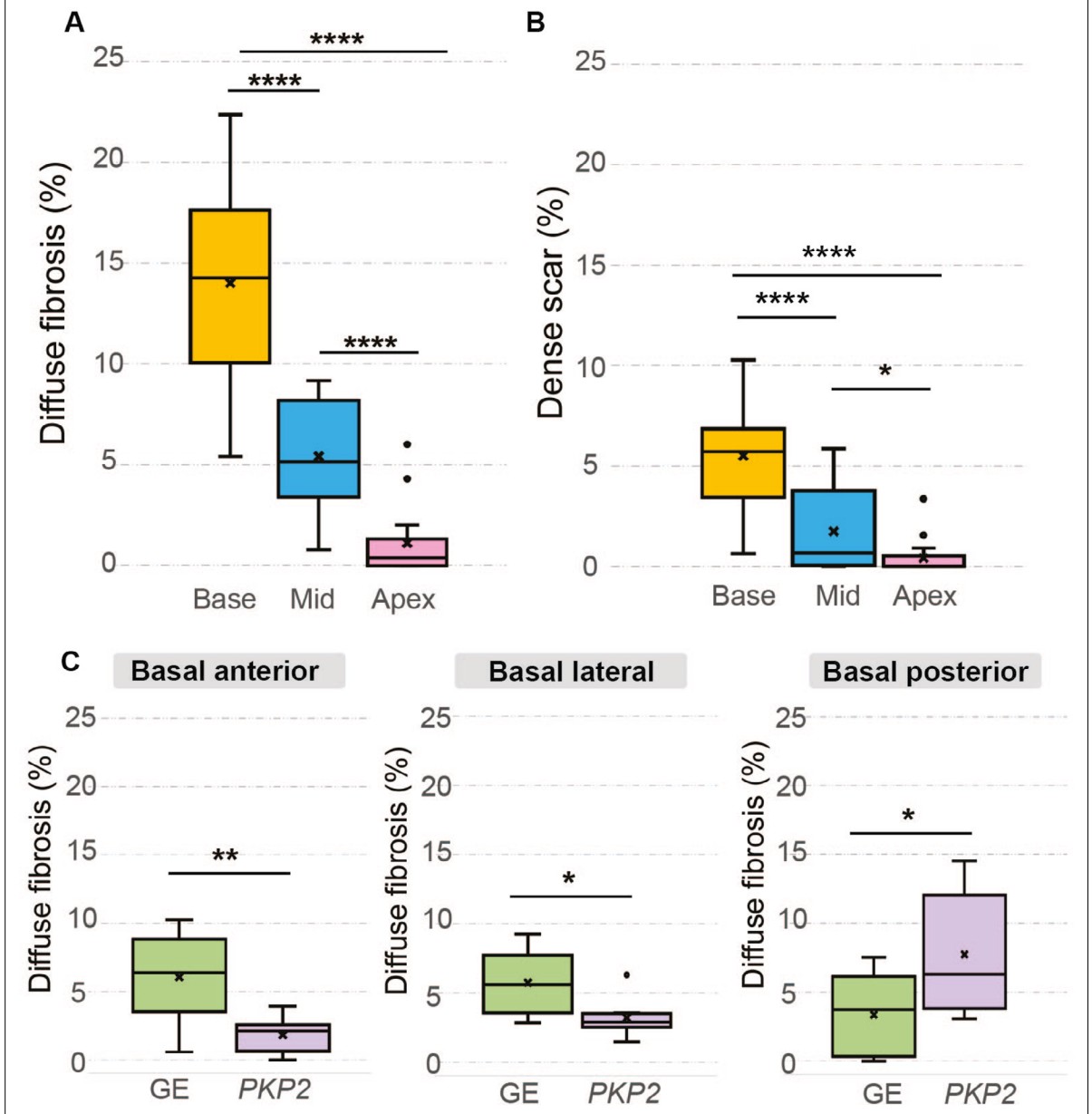

**Figure 4.** Amounts of diffuse fibrosis and dense scar in different regions of the RV and in the two patient groups. (**A, B**) Boxplots showing the amount of diffuse fibrosis (**A**, Base: $n = 16$, interquartile range [IQR] = 7.43; Mid: $n = 16$, IQR = 4.65; Apex: $n = 16$, IQR = 1.18; ****$p < 0.0001$) and dense scar (**B**, Base: $n = 16$, IQR = 2.73; Mid: $n = 16$, IQR = 3.14; Apex: $n = 16$, IQR = 0.36; *$p < 0.05$, ****$p < 0.0001$) in the entire ARVC cohort in different regions of the RV. (**C**) Boxplots comparing diffuse fibrosis amounts in different RV AHA segments between the two genotype groups. Significance was found in the following basal segments: basal anterior (GE: $n = 8$, IQR = 2.91; *PKP2*: $n = 8$, IQR = 1.78; **$p < 0.01$), basal lateral (GE: $n = 8$, IQR = 3.21; *PKP2*: $n = 8$, IQR = 0.75; *$p < 0.05$), and basal posterior (GE: $n = 8$, IQR = 4.74; *PKP2*: $n = 8$, IQR = 7.25; *$p < 0.05$). Amounts of diffuse fibrosis/dense scar are normalized with respect to the patient's total RV tissue volume. Median and interquartile range were represented in each boxplot, where dots indicate outliers. Paired Student's *t*-tests were applied to assess statistical significance.

for each RV AHA segment in all 16 ARVC patients. In the GE patient group, Geno-DT resulted in 28 distinct VTs being induced, 25 of which were observed during EPS. The *PKP2* group showed comparable results, with 25 VT morphologies induced in *PKP2* ARVC models and 25 reported in the EPS record. In terms of VT circuit locations, Geno-DT captured all VT locations that were observed during EPS except for one VT on the mid lateral RV wall in the GE group. In the *PKP2* group, Geno-DT predicted all the VT locations recorded during EPS, except for one VT on the basal anterior RV wall

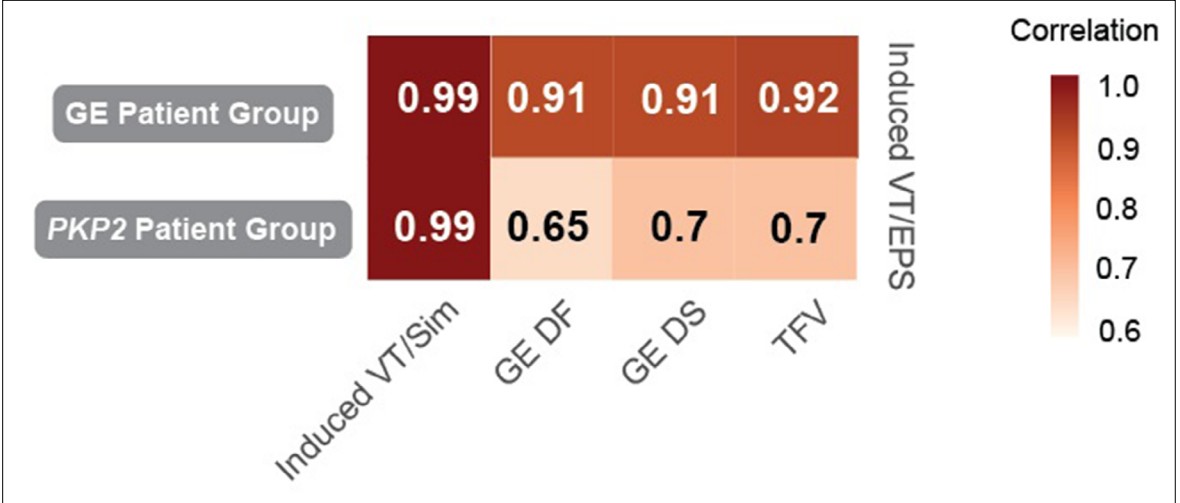

**Figure 5.** Correlation between the number of VTs induced during the EPS and structural remodeling features across different RV anatomical locations in ARVC patients. The number of VT episodes induced during EPS is highly correlated with the volume of structural remodeling (DF, DS, TFV) in GE patient group (top row), while correlated less in *PKP2* patient group (bottom row). The VT episodes induced during EPS and by Geno-DT are highly correlated in both patient groups (first column). Numbers in the block stand for the correlation coefficient (**r**) between the two corresponding variables. *r* > 0.7 stands for high correlation; 0.5 < r ≤ 0.7 stands for moderate correlation. Induced VT/EPS: VTs induced during the clinical EPS; Induced VT/Sim: VTs induced in Geno-DTs; DF: diffuse fibrosis; DS: dense scar; VT: ventricular tachycardia, TFV: total fibrotic volume = DF + DS.

and two on the basal posterior wall. Hence, in both cases, Geno-DT accurately predicted the locations of nearly all the VTs induced during EPS.

Additionally, for each individual ARVC patient, we compared the AHA segment of every VT circuit induced in each Geno-DT to the patient's EPS VT record obtained from EPS. In *Table 2*, we summarized the capability of Geno-DT in predicting VT locations. The results showed outstanding accuracy, sensitivity, and specificity in both GE and *PKP2* patient groups.

*Figure 7A and B* illustrate the above findings with two examples, one from GE and one from *PKP2* group. For the GE patient (*Figure 7A*), the genotype-matched model induced three VT morphologies: two figure-of-eights on the anterolateral and posterolateral RV wall and a single-reentry VT on the posterior RV wall. This GE patient had two VT circuits observed during the clinical EPS, each one on the basal and mid lateral wall of the patient's RV. Although the clinical EPS report did not mention VT induced on the posterior RV wall, it did mention that fractionated potentials, which are considered as VT circuit indicators, were observed in the posterior RV, and thus, clinical ablation was done there. Therefore, we consider all our predicted VTs in this patient in agreement with the clinical findings. Similarly, for the *PKP2* patient, the genotype-matched model induced five VTs: a figure-of-eight reentry on the anterolateral wall, a single-reentry VT and two figure-of-eight VTs on the lateral RV wall, as well a single-reentry VT on the posterior RV wall (*Figure 7B*). This *PKP2* patient underwent two clinical ablation procedures within one year. The first ablation identified three VTs, one in the mid lateral region, one in the basal posterolateral region and one in the basal posterior region. The EPS procedure during the second ablation identified two sustained VTs on the anterolateral and lateral wall. These examples showcase the excellent correspondence between Geno-DT VT circuit predictions and clinical observations.

## The role of genotype-specific EP remodeling in VT propensity

Once we ascertained the ability of Geno-DT to correctly predict VT circuits in ARVC patients, we investigated the contribution of pathogenic variants (via the corresponding genotype-specific EP properties) in creating propensity to VT. To do so, we switched the cellular models between GE and *PKP2* patient groups to introduce a mismatch between personalized structural remodeling and genotype-specific EP properties. We then repeated the simulations under these mismatched conditions. *Figure 8C* provides a summary of the findings. We also present examples in *Figure 8A and B*, which are the same two patients' ventricular models included in *Figure 7* but simulated under genotype-mismatched conditions. Evident from the figure is that VT reentrant circuits were also

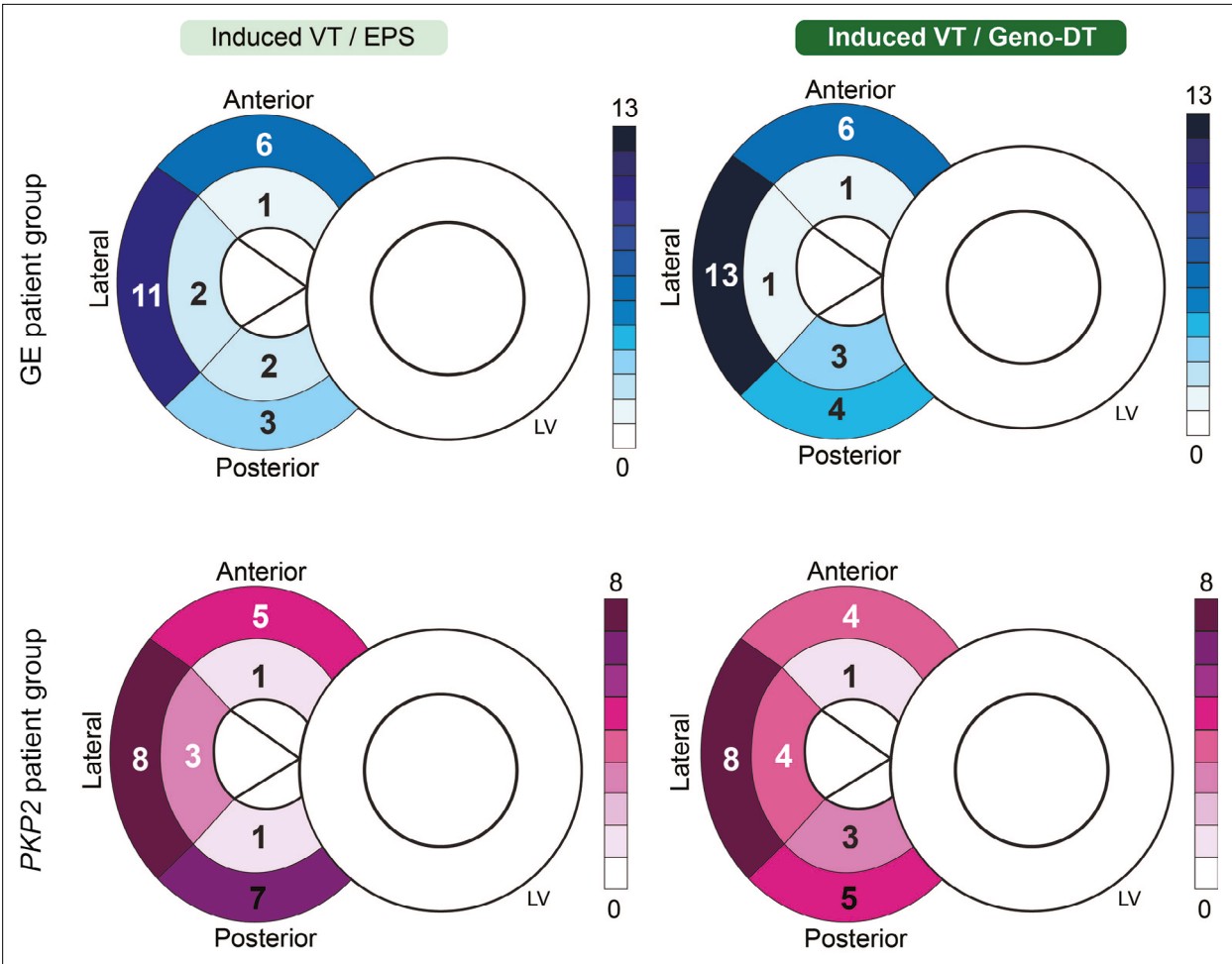

**Figure 6.** Comparison of the number of unique VT morphologies induced in Geno-DTs to VTs induced during EPS in each RV region. Schematics of the heart were labeled with the numbers of unique VT morphologies at different AHA segments on the RV induced during clinical EPS (left) and induced in Geno-DTs (right) of the two ARVC genotypes (GE: blue, *PKP2*: pink).

induced under genotype-mismatched conditions, however, they are distinctly different from those in the genotype-matched models, and very different from the VTs induced during EPS. In *Figure 8A*, the mismatched GE DT had only one VT circuit induced on the posterolateral RV wall, and in *Figure 8B* the mismatched *PKP2* DT presented only one VT in the anterolateral RV wall. For both the GE and *PKP2* examples, the genotype-mismatched DTs missed many VT circuits induced during EPS. The bullseye plots in *Figure 8C* emphasize the underprediction of the genotype-mismatched Geno-DTs, in comparison to those in *Figure 6*.

**Table 2.** Capability of Geno-DTs to predict VT locations.

|  | GE patient group (*n* = 8) | *PKP2* patient group (*n* = 8) |
|---|---|---|
| Sensitivity | 1.00 | 0.86 |
| Specificity | 0.94 | 0.90 |
| Accuracy | 0.96 | 0.89 |
| Error rate | 0.042 | 0.11 |
| F1 score | 0.89 | 0.83 |
| MCC | 0.90 | 0.76 |

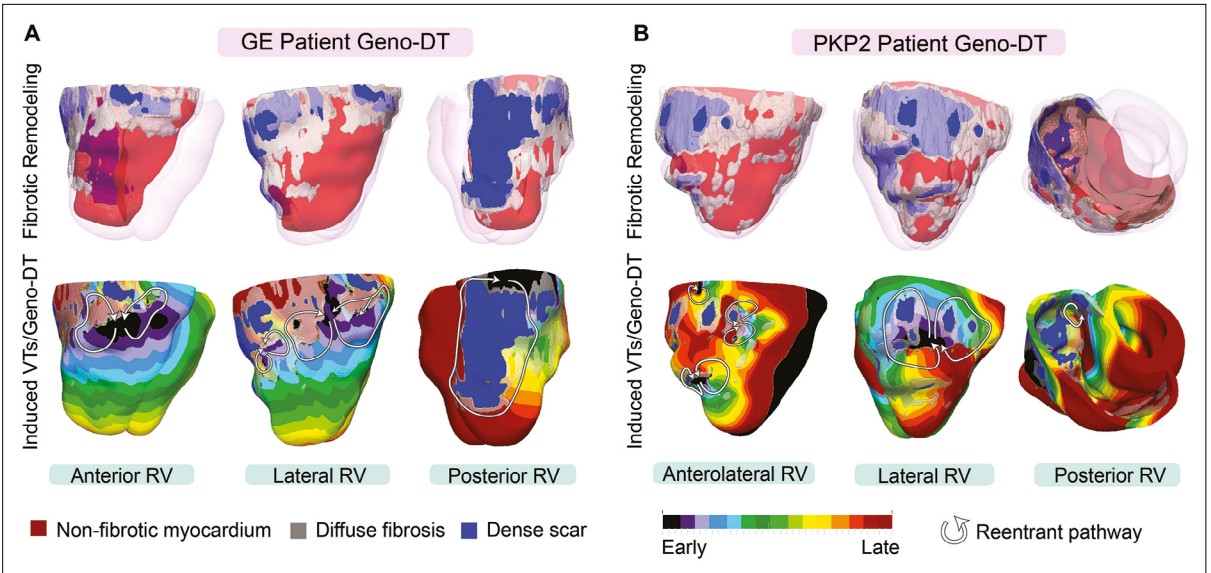

**Figure 7.** VTs induced by Geno-DT in two ARVC patients each from the GE and *PKP2* genotype groups. (**A, B**) The top row shows three different views of reconstructed geometrical models of GE and *PKP2* patient hearts with personalized diffuse fibrosis (gray) and dense scar (blue). The bottom row shows the activation patterns of the VT reentrant circuits induced by Geno-DT rapid pacing.

## Mechanisms by which genotype-specific EP remodeling alters VT circuits

To gain a better understanding of the mechanisms underlying the induction of different reentrant circuits resulting from EP remodeling, we analyzed in detail the VT circuits simulated under genotype-matched and genotype-mismatched conditions. Two examples of this analysis are presented in *Figure 9* for the *PKP2* ventricular model from *Figure 7B*. Animation of the VT circuits propagation is available in *Figure 9—video 1*.

Reentrant circuit 1 is the VT induced in the lateral RV wall of the *PKP2* (genotype-matched) DT (*Figure 9A*). This circuit involved two dense scar islands surrounded by diffuse fibrosis, creating an isthmus of low conductivity in between. The initial wavefront elicited from a basal posterior pacing site propagated around the dense scar and through the isthmus. As the exiting wavefront encountered non-fibrotic myocardium, it had a slow conduction due to the lowered upstroke velocity and elevated resting membrane potential resulting from *PKP2* EP remodeling. Consequently, downstream tissue failed to be excited due to source-sink mismatch. Two other wavefronts propagated from either side of the dense scar islands into this unexcited myocardium and collided. In the time it took for the two opposing wavefronts to meet, the non-fibrotic myocardium at the entrance of the isthmus remained refractory (*PKP2* cell model has extended refractoriness), thereby allowing the collided wavefront to travel back up the isthmus, meet refractory tissue and ultimately form the figure-of-eight reentry. In contrast, no reentry was induced at this same location with the incorporation of GE EP properties (*Figure 9B*). Specifically, as the wavefront left the isthmus and encountered non-fibrotic myocardium, its propagation speed was higher due to the GE cell model's rapid upstroke velocity and unchanged resting membrane potential. Because of the fast conduction velocity, the isthmus did not have time to recover, thereby excitation occurred only of tissue ahead of the wavefront.

Reentrant circuit 2 was induced in the posterior RV wall of the *PKP2* genotype-matched DT (*Figure 9C*). The wavefront elicited from the same basal posterior pacing site had slowed down significantly while traveling through a band of diffuse fibrosis. In areas where the band was wide, the conduction velocity decreased to the point where the wavefront was unable to excite non-fibrotic myocardium due to source-sink mismatch. This was a result of the fact that the *PKP2* cells have slowed upstroke velocity and elevated resting membrane potential. An adjacent wavefront was thereby able to turn into this unexcited region and initiate reentry. In contrast, no reentry occurred at this location in the GE DT (*Figure 9D*). The GE cell's rapid upstroke velocity allowed the wavefront to excite the

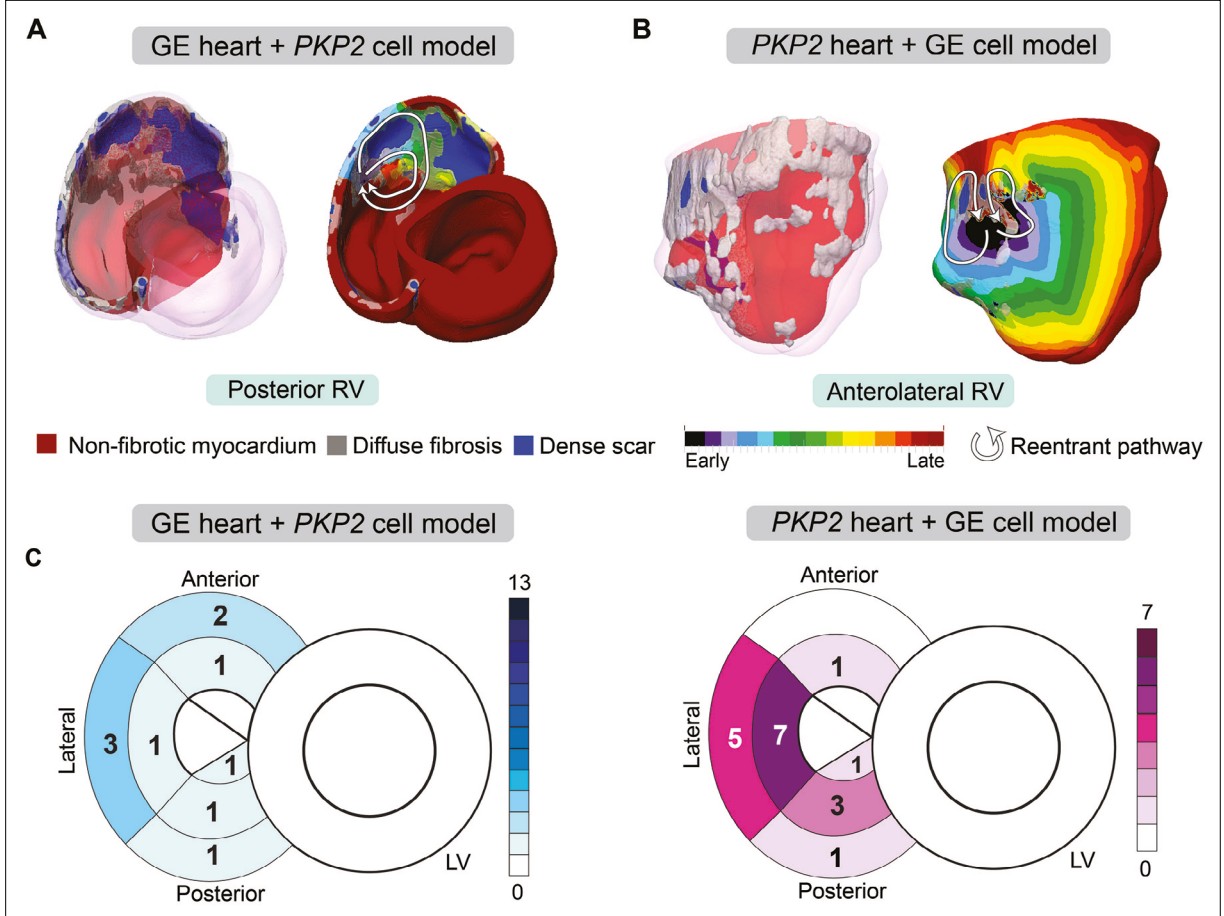

**Figure 8.** Genotype-mismatched DTs significantly underpredicted VTs in ARVC patients. (**A, B**) Geometrical ventricular models with structural substrates visualized (left); VT circuits induced in genotype-mismatched DTs (right). (**C**) Bullseye plots labeled with the number of unique VT morphologies in different RV AHA segments induced in Geno-DTs under genotype-mismatched conditions.

non-fibrotic myocardium after exiting the wide diffuse fibrosis band. Consequently, the wavefront leaving the diffuse fibrosis was more planar, avoiding source-sink mismatch, and did not result in reentry.

Together, these results help illustrate the contribution of genotype-based EP differences to the variability of organ-level wavefront propagation patterns.

## Discussion

This study presents a novel heart digital-twin approach called Geno-DT, tailored to predict VT reentrant circuits in patients with different ARVC genotypes. We demonstrated that Geno-DT has excellent capability in predicting VT reentrant circuits non-invasively for patients with both gene-elusive and *PKP2* positive genotypes. In predicting VT circuits, the study also reveals new mechanistic insights regarding VT arrhythmogenesis in ARVC. By comparing the VT circuits predicted by Geno-DT to the clinical EPS observations, we demonstrated that Geno-DT has high sensitivity, specificity, and accuracy in predicting non-invasively VT circuits and their locations in both GE and *PKP2* genotypes (100%, 94%, 96% in GE patient group; 86%, 90%, 89% in *PKP2* patient group). The Geno-DT approach thus has the potential to improve pre-ablation planning and to lead to tailored personalized ablation strategies. Our study further charts a pathway towards bringing computational modeling in clinical decision-making thus augmenting precision medicine approaches in cardiology.

With its incorporation of genetic EP information, Geno-DT is a novel development in heart digital twin applications. Although previous studies have included cellular remodeling in predicting VT risk for non-ischemic cardiac diseases such as hypertrophic cardiomyopathy and Tetralogy of Fallot (*Shade*

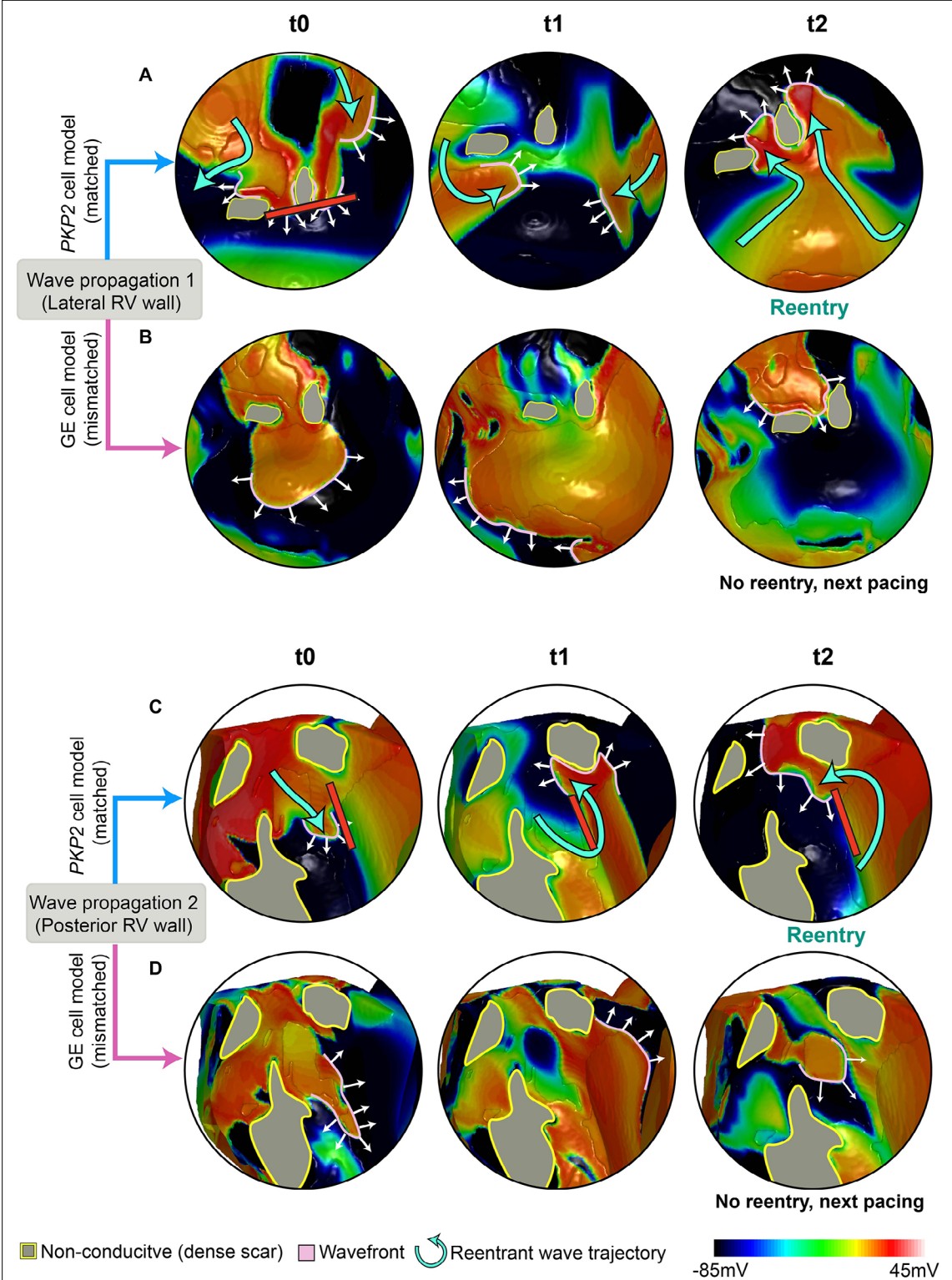

**Figure 9.** Comparison of propagation following pacing in a *PKP2* patient's ventricular model using different cell models. Each row of the figures shows a series of frames that depict the continuous wave propagation in a portion of the *PKP2* patient's DT from ***Figure 7B*** (genotype-matched conditions, [A] and [C]) and ***Figure 8B*** (genotype-mismatched conditions, [B] and [D]). The images in each column (**A and B, C and D**) show the same time instant. Pink lines mark the wavefronts, white arrows indicate direction of propagation and cyan curved arrows in (**A**) and (**C**) represent reentrant wave trajectories. Gray areas contoured by yellow represent the dense scar (DS) regions that are non-conductive. (**A, B**) (**A**) and (**B**) compare the wave propagation on the lateral RV wall between genotype-matched and -mismatched conditions. At t0, the wavefront in (**A**) propagates slower than that in (**B**), resulting in

*Figure 9 continued on next page*

*Figure 9 continued*

conduction block in (**A**) but not in (**B**). In (**A**), following wavefront fusion at t1, reentry is established at t2. In (**B**), no reentry forms and the next pacing stimulus captures. (**C, D**) (**C**) and (**D**) compare the wave propagation on posterior RV wall between genotype-matched and -mismatched conditions. At t0, the wavefront in (**C**) propagates slower than that in (**D**). Due to the extended refractoriness in *PKP2*, tissue adjacent to the wavefront is still recovering, and conduction block takes place (red line). In (**D**), tissue ahead of the wavefront is fully recovered, allowing it to propagate through. At t1, in (**C**), the wavefront, with a high curvature, travels around the conduction block. In (**D**), the wavefront is more planar and continues to propagate. At t2, a reentrant circuit forms in (**C**); while in (**D**), no reentry forms and the next pacing stimulus captures.

The online version of this article includes the following video for figure 9:

**Figure 9—video 1.** Animation of two VT circuits propagation corresponding to *Figure 9*.

https://elifesciences.org/articles/88865/figures#fig9video1

*et al., 2020*; *O'Hara et al., 2022*), personal genetics were never considered. ARVC is associated with several genotypes, each causing specific pathological alterations to ion channels (*Ohno, 2016*). However, it has remained unclear how the altered cellular EP is reflected at the organ and how it affects the likelihood of VT occurrence. Our study addresses these questions by developing a *PKP2* cell model that incorporates the EP effects of pathogenic variants and is compatible with whole-heart modeling. While a previous study had incorporated *PKP2* cell EP properties in a computational cell model (*Lyon et al., 2021*), the model was too complex to be incorporated into whole-heart simulations due to its focus on describing subcellular phenomena.

Our study offers new insights into the mechanisms underlying VT propensity in patients with ARVC. While previous clinical observations have suggested that sustained VTs are strongly associated with fibrotic remodeling in the basal anterior RV, it has been unclear whether this association holds true between the GE and *PKP2* genotypes (*Basso et al., 2008*; *Corrado et al., 2005*). Studies of ARVC structural substrates have typically relied on electroanatomical mapping and biopsy, which do not provide a complete characterization of the 3D fibrosis distribution in the right ventricle (*Te Riele et al., 2013*; *Basso et al., 2009*; *Corrado et al., 2005*). Here, we reconstructed heart digital twins using LGE-CMR to quantify the extent and distribution of fibrotic remodeling in each patient's heart. We found that *PKP2* patients with clinical sustained VTs had more fibrotic remodeling in the basal posterior region of RV than the anterior, while the fibrosis distribution in the GE group was consistent with previous findings. These results demonstrate that different ARVC genotypes exhibit distinct distributions of fibrotic remodeling. Furthermore, while the number of induced VT morphologies was similar between GE and *PKP2* patients, our analysis revealed a weaker correlation between the locations of VTs induced during EPS and distribution of fibrosis in the *PKP2* group compared to the GE group. This suggests that, in addition to structural remodeling, unique *PKP2*-specific changes in cellular EP properties may contribute to the development of VT in *PKP2* patients.

Our findings further elucidate the important role of genotype-specific EP remodeling in sustaining VT circuits in ARVC. Previous study has shown that a decrease in $V_{max}$ directly affects the conduction velocity in the cardiac tissue (*Issa et al., 2019*). Consistent with this relationship, we observed that the reduced $V_{max}$ in the *PKP2* cell model, caused by downregulated $I_{Na}$, resulted in slower wavefront conduction, particularly in non-fibrotic regions adjacent to the dense fibrosis. The different restitution properties of the *PKP2* cell model resulting from the altered calcium handling also contributed to the formation of reentry. The rapid pacing rates used to induce VT corresponded to the steep initial phase of the restitution curves, where *PKP2* cell model had extended refractoriness compared to the GE cell. Hence in *PKP2* hearts, extended refractoriness and decreased conduction velocity resulted in wavefronts of higher curvature, subsequent source-sink mismatch and conduction block, setting the stage for arrhythmogenesis.

We identify several potential benefits of using Geno-DT in ARVC clinical management. Firstly, it could save time during the ablation procedure as all possible locations that sustain VTs could be known prior to the procedure. Secondly, it provides a new approach for substrate modification in ARVC ablation, in which the substrate is targeted at predicted VT locations instead of ablating fibrotic tissue or all regions with voltage abnormalities on electroanatomical mapping, as not all of these are arrhythmogenic (*Rottmann et al., 2019*). The latter ablation approaches could lead to excessive lesions, which have negative consequences on cardiac function, especially in ARVC, as it is characterized by abnormal inflammation response (*Asatryan et al., 2021*; *Briceño et al., 2020*; *Santangeli et al., 2015*). Lastly, our study demonstrated that the underlying mechanisms of sustaining VT

reentrant circuits differ among various ARVC genotypes. This finding underscores the importance of genotype-specific ARVC management in clinical practice, which can be facilitated by the Geno-DT approach.

The success of our Geno-DT approach in predicting VT circuit locations underscores its potential for translation to the clinical setting. As the understanding of human genetic variation underlying cardiovascular diseases continues to evolve with advances in functional genomics (*Li et al., 2022*), Geno-DT could provide a generalized framework to integrate multimodal clinical data and improve precision health for each individual patient.

## Materials and methods

### Patient population

This retrospective study cohort included 16 patients from the Johns Hopkins ARVC database who were diagnosed with ARVC based on 2010 Task Force Criteria between 2010 and 2018. Patients were included if they (1) underwent genetic testing for ARVC risk variants that demonstrated either no pathogenic variants (GE patients) or pathogenic *PKP2* variant (*PKP2* patients), (2) had inducible VT during EPS, and (3) had LGE-CMR demonstrating the presence of RV fibrosis. Clinical reports from EPS were reviewed to identify both the number of distinct VT morphologies and the locations of induced VT within the RV. Patients' CMR images were reviewed clinically to identify the presence/absence of RV enhancement (S.Z.). Patient clinical characteristics are shown in *Table 1*.

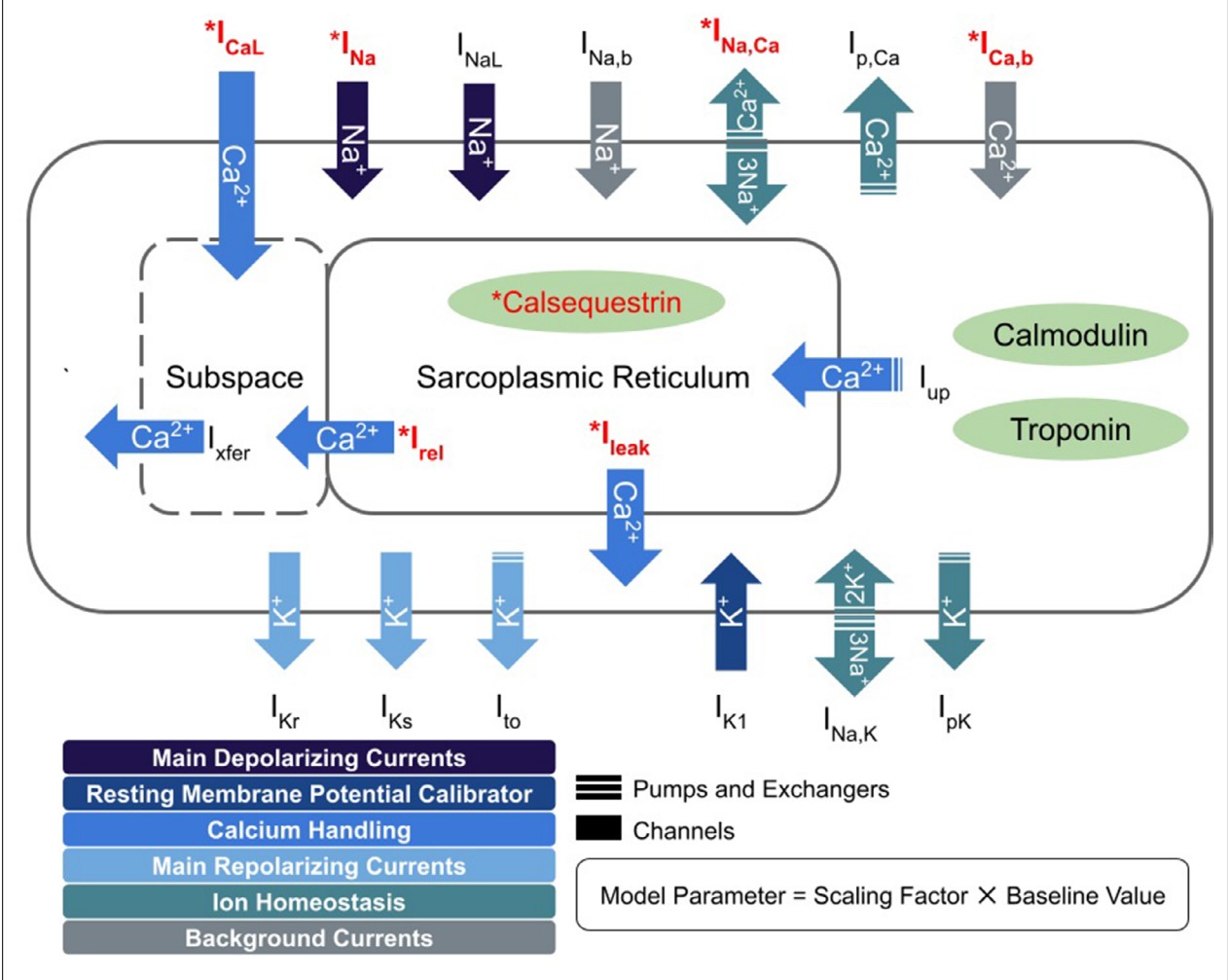

**Figure 10.** Schematic diagram describing the ionic currents across the cell membrane and sarcoplasmic reticulum of an adult human cardiomyocyte. Components with asterisks (red color) were modified from their baseline formulations to reflect *PKP2* pathogenic ionic remodeling.

## Cell-level modeling

The different EP properties of cardiomyocytes in the GE and *PKP2* genotype groups were represented with two different cell ionic models. For non-fibrotic myocardium regions of the GE group, we used the *ten Tusscher and Panfilov, 2006* model with the addition of a late sodium current representation (*O'Hara et al., 2011*). A late sodium current ($I_{NaL}$) formulation was added, as done in our previous studies (*Cartoski et al., 2019*; *Shade et al., 2020*; *Arevalo et al., 2016*; *Prakosa et al., 2018*; *Shade et al., 2021*). For non-fibrotic myocardium regions of the *PKP2* group, we developed a new *PKP2* model by modifying the GE model based on experimental data (*Lyon et al., 2021* and *Sato et al., 2009*). These modifications are summarized in *Figure 10*. Our resulting *PKP2* model recapitulates the altered sodium current dynamics and calcium-handling properties observed in experimental recordings of cardiomyocytes from ARVC hearts.

In our *PKP2* ionic model, the maximum channel conductance for sodium current ($I_{Na}$) was decreased by 70% to reflect reported experimental current dynamics from *Sato et al., 2009*. The calcium-handling in our *PKP2* model was adjusted to reproduce experimental data from *Lyon et al., 2021*, which reported a decreased L-type calcium current ($I_{CaL}$) and higher amplitude calcium transient (CaT). We scaled the maximal conductance of $I_{CaL}$ by 50%, reducing its peak current (9.03 vs 6.84 pA/pF) proportional to the experimental values. To counteract the accompanying decrease in CaT amplitude produced by this modification, we also upregulated the background calcium current ($I_{bCa}$), representative of connexin 43 (Cx43) hemichannel-mediated calcium entry, by fivefold. This upregulation was in line with the in vitro properties of PKP2-deficient cells: disrupted cell-cell adhesion, functional dysregulation of Cx43, increased membrane calcium permeability, and excess free calcium concentrations.

Studies from Lyon et al. have further shown that *PKP2*-deficient cardiomyocytes have reduced ryanodine receptor (RYR2) channel expression but enhanced channel sensitivity to calcium. We incorporated reduced channel expression as decreases in maximal rate of calcium movement for both the RyR2-mediated SR release (0.102 vs 0.0816 ms-1) and leak (3.61E-4 vs 2.88E-4 ms-1) currents. Since the baseline TT2 model did not represent individual RyR2 channels, we indirectly accounted for increased sensitivity by increasing free calcium concentrations in the SR (CaSR) and subspace (CaSS). CaSR overload was achieved by decreasing calsequestrin affinity as a 40% reduction in the half-saturation constant for sarcoplasmic buffering (0.3 vs 0.18 mM). CaSS was increased from the reduced calcium diffusion gradient between the subspace and the cytosol; a 20% downregulation of the sodium-calcium exchanger current ($I_{NaCa}$) further contributed to this reduced gradient. The enhanced calcium concentration of all three compartments corresponded with the experimentally measured calcium levels for *PKP2*-deficient cardiomyocytes and effectively shifted the concentration vs RyR2-binding curve leftward.

To create the action potential duration (APD) restitution curves of the GE and *PKP2* models, we used a standard S1-S2 stimulus protocol similar to that described in *ten Tusscher and Panfilov, 2006*. Twenty S1 stimuli were applied at a basic cycle length (BCL) of 1 Hz to allow the models to reach a steady state. A single S2 extra stimulus of twice the diastolic threshold was then delivered at some diastolic interval (DI) following the action potential generated by the last S1 stimulus. The model was allowed to retain steady state (5 beats at BCL) before another S2 stimulus was applied at a decreased DI. This step was repeated for DIs decreasing from 1000 ms to 50ms in intervals of 25ms. An APD restitution curve was generated by plotting the APDs at 90% repolarization generated by the S2 stimuli against preceding DIs.

## EP properties for regions with structural remodeling

For regions of diffuse fibrosis, we used the TT2 model with modifications based on data reported by *Coppini et al., 2013*. The EP properties of this modified model have been validated in previous studies (*Shade et al., 2020*; *O'Hara et al., 2022*). Maximal conductance of $I_{NaL}$ and $I_{CaL}$ were respectively increased by 107% and 19%; maximal conductance of $I_{Kr}$, $I_{Ks}$, $I_{to}$, and $I_{K1}$ were respectively decreased by 34%, 27%, 85%, and 15%; sodium-calcium exchanger activity was upregulated by 34%; sarcoplasmic/endoplasmic reticulum calcium ATPase activity was downregulated by 43%.

## Geometrical model reconstruction

Geometrical model of each patient's ventricles was reconstructed based on the patient's 2D LGE-CMR scans. Each diastolic short-axis image stack had a median of 11 slices with an average of 9.58±2.16 mm

slice thickness and average in-plane axial resolution of 1.56±0.49 mm. Both the RV and LV myocardium were segmented from the LGE-CMR using a semi-automatic function of CardioViz3D which has been previously validated by our team (*Arevalo et al., 2016*; *Shade et al., 2020*; *Cartoski et al., 2019*). This segmentation method utilized landmarks to define the endocardial and epicardial boundaries.

Unlike our previous digital twin studies in ischemic cardiomyopathy (*Arevalo et al., 2016*; *Shade et al., 2020*; *Cartoski et al., 2019*), patients in the current cohort were right-dominant ARVC patients with only RV fibrosis detectable on LGE-MRI. We used Otsu thresholding to binarize the segmented myocardium into high- and low-intensity regions. The mean value and the standard deviation (SD) of the low-intensity region was calculated. We used the mean value of the low-intensity region as the reference mean of non-fibrotic myocardium and an intensity of ≥4 SD above the reference mean was classified as the dense scar region. Voxels between 2 and 4 SDs above the reference mean were classified as diffuse fibrosis; all other voxels were labeled as non-fibrotic myocardium (*Zhang et al., 2021*).

After identifying all different tissue types, we generated high-resolution finite-element tetrahedral meshes with an average resolution of 398 μm from the segmentation using finite-element analysis software (Mimics Innovation Suite; Materialise, Leuven, Belgium), as done previously (*O'Hara et al., 2022*). We then used a validated rule-based method to incorporate fiber orientations to each element of the computational mesh (*Bayer et al., 2012*).

## Simulation protocol and VT assessment

We used the openCARP software package to simulate electrical activity in monodomain representation of the myocardium (*Plank et al., 2021*). Full details regarding the simulation of electrical activity in the heart digital twins can be found in previous publications (*O'Hara et al., 2022*; *Prakosa et al., 2018*; *Sung et al., 2020*; *Shade et al., 2020*). Each heart digital twin was paced sequentially from 9 uniformly distributed endocardial RV locations (see *Figure 1*) using a validated rapid pacing protocol described in our previous studies (*Prakosa et al., 2018*; *Arevalo et al., 2016*; *Cartoski et al., 2019*). At each pacing site, six pacing stimuli (S1) were delivered at a 600ms cycle length, followed by a premature stimulus (S2) delivered 300ms after S1. If S2 did not result in reentrant arrhythmia, we shortened the S1-S2 interval in 10ms steps until arrhythmia was induced or the S2 failed to capture the tissue. If arrhythmia was not induced, we delivered additional stimuli (S3 and S4) in the same way as S2. Induced VT in the Geno-DT models was defined as re-entry after sustaining at least 2 cycles at the same critical site, as in our previous studies (*Sung et al., 2020*; *Shade et al., 2021*).

After inducing the re-entries, we analyzed activation maps to identify the VT morphologies and locations. Reentries induced from different pacing sites but occurring in the same location with the same morphology were classified as repetitive. Only unique VT circuits were counted. Simulations were conducted blind to the clinical data.

## Study limitations

Our study has a small sample size, limited by the fact that ARVC is relatively a rare disease. Moreover, since the goal of this study was to develop an image-based heart digital twin approach to accurately predict VT circuit locations, LGE-CMR scans and EPS records were needed to reconstruct the model and validate our predictions respectively; only a few patients in the database had both. Additionally, our study focused on patients with only RV enhancement since research has shown that most clinical VT ablation sites are on the RV for ARVC, even for those with LV involvement (*Marchlinski et al., 2021*). Furthermore, we considered only two genotypes, GE and *PKP2*, as human experimental data on cellular EP properties for other pathogenic variants implicated in ARVC (e.g., *DSP, DSG-2*), from which to construct cell models, is lacking; in addition, because the prevalence of these causal variants is lower compared to that of *PKP2*, there were not enough patients in the database with the corresponding clinical data needed for model construction and validation (clinical images, EPS study, etc.).

## Acknowledgements

The Johns Hopkins ARVD/C Program is supported by the Leonie-Wild Foundation, the Leyla Erkan Family Fund for ARVD Research, The Hugh Calkins, Marvin H Weiner, and Jacqueline J Bernstein Cardiac Arrhythmia Center, the Dr. Francis P Chiramonte Private Foundation, the Dr. Satish, Rupal, and Robin Shah ARVD Fund at Johns Hopkins, the Bogle Foundation, the Campanella family, the

Patrick J Harrison Family, the Peter French Memorial Foundation, and the Wilmerding Endowments. We are grateful to the ARVC patients and families who have made this work possible.

## Additional information

### Funding

| Funder | Grant reference number | Author |
|---|---|---|
| National Institutes of Health | R01-HL142496 | Natalia A Trayanova |
| National Institutes of Health | R01HL126802 | Natalia A Trayanova |
| Johns Hopkins University | Discovery Award | Adityo Prakosa |
| National Institutes of Health | T32HL007227 | Richard Carrick |
| National Institutes of Health | L30HL165535 | Richard Carrick |

The funders had no role in study design, data collection and interpretation, or the decision to submit the work for publication.

### Author contributions

Yingnan Zhang, Conceptualization, Resources, Data curation, Software, Formal analysis, Validation, Investigation, Visualization, Methodology, Writing – original draft, Project administration, Writing – review and editing; Kelly Zhang, Software, Formal analysis, Visualization, Methodology, Writing – original draft, Writing – review and editing; Adityo Prakosa, Software, Funding acquisition, Investigation, Methodology, Writing – review and editing; Cynthia James, Crystal Tichnell, Brittney Murray, Resources, Data curation, Project administration, Writing – review and editing; Stefan L Zimmerman, Resources, Data curation, Methodology, Writing – original draft, Writing – review and editing; Richard Carrick, Resources, Data curation, Funding acquisition, Methodology, Writing – original draft, Writing – review and editing; Eric Sung, Conceptualization, Software, Validation, Methodology, Writing – review and editing; Alessio Gasperetti, Resources, Data curation, Writing – review and editing; Hugh Calkins, Conceptualization, Resources, Data curation, Supervision, Funding acquisition, Project administration, Writing – review and editing; Natalia A Trayanova, Conceptualization, Resources, Data curation, Supervision, Funding acquisition, Investigation, Methodology, Writing – original draft, Project administration, Writing – review and editing

### Author ORCIDs

Yingnan Zhang ![ORCID] https://orcid.org/0000-0002-2070-6786
Kelly Zhang ![ORCID] http://orcid.org/0000-0001-8666-4569
Adityo Prakosa ![ORCID] http://orcid.org/0000-0002-1590-0322
Natalia A Trayanova ![ORCID] http://orcid.org/0000-0002-8661-063X

### Ethics

This retrospective study was approved by Johns Hopkins Medicine Institutional Review Board (IRB) #1, IRB title: Clinical and genetic investigations of right ventricular dysplasia; IRB identifier: NA_00041248; IRB PI: Hugh Calkins. All participants provided written informed consent.

Reviewer #1 (Public Review): https://doi.org/10.7554/eLife.88865.3.sa1
Reviewer #2 (Public Review): https://doi.org/10.7554/eLife.88865.3.sa2
Reviewer #3 (Public Review): https://doi.org/10.7554/eLife.88865.3.sa3
Author Response https://doi.org/10.7554/eLife.88865.3.sa4

## Additional files

### Supplementary files
• MDAR checklist

### Data availability

Patient-derived data including CMR images are not publicly available to respect patient privacy. Interested parties wishing to obtain these data for non-commercial reuse should contact the corresponding author via email; the request will need to be approved by IRB. The image processing software Cardio-Viz3D can be freely obtained from http://www-sop.inria.fr/asclepios/software/CardioViz3D. The cell models are freely available from the repository CellML (https://models.cellml.org/exposure/de5058f16f829f91a1e4e5990a10ed71). Documentation and instructions on the use of the openCARP cardiac electrophysiology simulator and meshalyzer visualization software are available via https://opencarp.org/.

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
