## [Editor Report · eLife assessment]

This **important** study brings together a clear application of the digital twin approach to make predictions using patient specific models with different genotypes. The data are **compelling** and go beyond the current state-of-the-art to support proof-of-principle evidence. Given the low subject numbers, further studies will be required going forward to support the veracity of the data and its translational utility.

---

## [Referee Report · Reviewer #1 (Public Review)]

The authors have employed a digital twin approach to show that depending on the underlying disease mechanism, a digital replica constructed from human data can both recapitulate clinical findings, but also provide important insights into the fundamental disease state by revealing underlying contributing mechanisms. Moreover, the authors are able to show that a disease state caused by two different underlying genetic anomalies exhibit different electrical and morphological profiles.

This is important information as it allows for potential stratification of treatment approaches in future cases based on underlying phenotype by linking it to specific genotype properties. One of the most innovative aspects of the study is the mismatch switching between personalized structure, remodeling and genotype specific electrophysiological properties. The approach is elegant and allows for further exposure of the key mechanisms that contribute to the development of ventricular tachycardia circuits. One addition that could add more insight is to predict the effect of structural remodeling alone well, considering only normal electrophysiological models. Another interesting approach would be a sensitivity analysis, to determine how sensitive the VT circuits are to the specific geometry of the patient and remodeling that occurs during the disease, such an approach could also be used to determine how sensitive the outputs are to electrophysiological model inputs.

---

## [Referee Report · Reviewer #2 (Public Review)]

The authors of this paper use a "digital twin" computational model of electrophysiology to investigate the pathology of Arrhythmogenic Right Ventricular Cardiomyopathy (ARVC) in several patients undergoing Electro-Physiological Studies (EPS) to treat Ventricular Tachycardias (VTs). The digital twin computational models are customised to the individual patient in two ways. Firstly, information on the patient's heart geometry and muscle/fibrous structure is extracted from Late Gadolium-Enhanced Magnetic Resonance Image (LGE-MRI) scans. Secondly, information from the patient's genotype is used to decide the particular electrophysiological cell model to use in the computational model. The two patient genotypes investigated include a Gene Ellusive (GE) group characterised by abnormal fibrous but normal cell electrical physiology and a palakophilin-2 (PKP2) group in which patients have abnormal fibrotic remodelling and distorted electrical conduction. The computational model predicts the locations and pathways of re-entrant circuits that cause VT. The model results are compared to previous recordings of induced VTs obtained from EPS studies.

The paper is very well written, and the modelling study is well thought out and thorough and represents an exemplar in the field. The major strengths of the paper are the use of a personalised patient model (geometry, fibrous structure and genotype) in a clinically relevant setting. Such a comprehensive personal model puts this paper at the forefront of such models in the field. The main weaknesses of the paper are more of a reflection on what is required for creating such models than on the study itself. As the authors acknowledge, the number of patients in each group is small. Additional patients would allow for statistical significance to be investigated.

The paper's authors set out to demonstrate the use of a "digital twin" computational model in the clinical setting of ARVC. The main findings of the paper were threefold. Firstly, the locations of VTs could be accurately predicted. There was a difference in the abnormal fibrous structure between the two genotype groups. Finally, there was an interplay between the fibrous structure of the heart and the cellular electrophysiology in that the fibrous remodelling was responsible for VTs in the GE group, but in the PKP2 group VTs were caused by slowed electrical conduction and altered restitution. The study successfully met the aims of the paper.

The major impact of the paper will be in demonstrating that a personalised computational model can (a) be developed from available measurements (albeit at the high end of what would normally be measured clinically) and (b) generate accurate results that may prove helpful in a clinical setting. Another impact is the finding in the paper that the cause of VTs may be different for the two genotypes investigated. The different interplay between fibrous and electrophysiology suggested by the modelling results may provide insights into different treatments for the different genotypes of the pathology. The authors use open-source software and have deposited all non-confidential data in publically available repositories.

---

## [Referee Report · Reviewer #3 (Public Review)]

Overview

The authors propose a personalized ventricular computational model (Geno-DT) that incorporates the patient's structural remodeling (fibrosis and scar locations based on LGE-CMR scans) as well as genotyping (cell membrane kinetics based on genetic testing results) to predict VT locations and morphologies in ARVC setting.

To test the model, the authors conducted a retrospective study using 16 ARVC patient data with two genotypes (PKP2, GE) and reported high degree of sensitivity, specificity, and accuracy. In addition, the authors determined that in GE patients, VT was driven by fibrotic remodeling, whereas, in PKP2 patients, VT was associated with a combination of structural and electrical remodeling (slowed conduction and altered restitution).

Based on the findings, the authors recommend using Geno-DT approach to augment therapeutic accuracy in treatment of ARVC patients.

Critiques

1. The small sample size is a limitation but has already been acknowledge and documented by the authors.

2. Another limitation is the consideration of only two of the possible genotypes in developing the cell membrane kinetics, but again has acknowledged by the authors.

Final Thoughts

The authors have done a commendable job in targeting a disease phenotype that is relatively rare, which constrains the type of data that can be collected for research. Their personalized computational model approach makes a valuable contribution in furthering our understanding of ARVC mechanisms.

---

## [Author Response]

We would like to express our gratitude to the Editors and Reviewers for their thoughtful and helpful comments. We sincerely appreciate the opportunity to submit our revised manuscript titled “Predicting Ventricular Tachycardia Circuits in Patients with Arrhythmogenic Right Ventricular Cardiomyopathy using Genotype-specific Heart Digital Twins” to eLife. We are delighted that our research in ARVC has garnered the interest of the three reviewers. Below, we provide our point-by-point responses to the reviewers’ comments. We have also incorporated the suggestions provided by the reviewers in our revised manuscript.

**Comments from Reviewer 1**

We thank Reviewer 1 for their positive assessment and thoughtful suggestions. Here are the responses to the comments of reviewer 1:

Comment 1: One addition that could add more insight is to predict the effect of structural remodeling alone well, considering only normal electrophysiological models.

We thank the reviewer to give this thoughtful suggestion to our experiment design. We would like to highlight that this suggestion was indeed taken into consideration in our study as all the patients’ hearts were modeled using the gene-elusive cell model before the structural-EP mismatch was implemented. The gene-elusive cell model is a baseline ten Tusscher (TT2) human ventricular model described in the “Cell-level modeling” of our Methods. Therefore, we have already examined the impact of structural remodeling alone in the study.

Comment 2: Another interesting approach would be a sensitivity analysis, to determine how sensitive the VT circuits are to the specific geometry of the patient and remodeling that occurs during the disease, such an approach could also be used to determine how sensitive the outputs are to electrophysiological model inputs.

We think this suggestion is of great value and could benefit our future ARVC studies. The reviewer pointed out the importance of investigating how sensitive the VT circuits are to the specific geometry/remodeling of the patient during disease progression. To achieve this, for each patient, a sequence of LGE-CMR images at different stages of this disease is required for model reconstruction; unfortunately, our cohort for this study does not incorporate such data.

Comments from Reviewer 2

We thank Reviewer 2 for the positive assessment, and here are the responses to the comments:

Comment 1: I appreciate that the types of computational models detailed in this paper take enormous time to develop. However, to identify bottlenecks in the clinical workflow (and thus targets for future research), it may be nice for the authors to discuss the time taken to generate and run the models for each patient?

We sincerely appreciate the valuable feedback from the reviewer. We recognize the importance of considering model generation and run time. In the introduction, we have highlighted the clinical challenge in managing ARVC ablation procedures, which is the inability to capture all the VT due to an incomplete understanding of VT mechanisms. We acknowledge the reviewer’s concern regarding the potential time taken by the model to predict VT circuits and whether this could hinder the integration into the current ablation procedure. However, it is important to clarify that our model is primarily based on clinical images obtained in advance of the procedure. As a result, there is sufficient time available to generate the results required for ablation planning.

Comment 2: In the Materials and Methods section, some references are underlined? Is this a typo or meant to convey some particular information?

We thank the reviewer for pointing this typo out and we have removed the underlining of references in our revised manuscript.

Comment 3: The authors state that the cellular models are available from the CellML model repository. This is an excellent practice. However, the URL that is given points to the entire CellML website. It will be more useful for URLs that point to the specific models used in the study so that readers can be sure they are looking at the correct model.

We appreciate the reviewer for this suggestion, and we have edited the URL in Data Availability to link to a specific cell model on the CellML website.

Comment 4: In the abstract, the authors report the sensitivity, specificity, and accuracy of their computer models but fail to comment in the abstract that they are comparing against recordings from the patient during a previous EPS study. To assist further readers who are scanning the abstract, the authors may wish to add a sentence or two to detail what they are comparing their model results to.

We thank the reviewer for the suggestion. This is a retrospective study. We recognize the importance of wording clarity in the abstract; in response, we have added a sentence in the abstract to clarify that we compared VT locations of Geno-DT with the ones recorded during clinical EPS to obtain sensitivity, specificity, and accuracy.

Comment 5: In Table 1 some of the data is discrete e.g., the number of patients on a beta-blocker. The authors give a p-value for comparing the GE and PKP2 data and state in the caption that a Student's t-test has been used. Strictly speaking, a t-test is not really appropriate for the population proportion with non-parametric data. That said, the size (n) of the data here makes the p-values from any statistic very unreliable. Perhaps the authors might like to reconsider if p-values add anything to such data? If so, then the statistical test should be reconsidered.

We truly appreciate the reviewer for pointing out this typo in the caption of Table 1. For the non-parametric discrete data, we used z-test, a common statistical method used to compare percentages, to get the p values, but we mistakenly only mentioned t-test in our caption. We acknowledge the limitation of our sample size and we have corrected this typo in our revision.

Comment 6: I found Table 1 and its caption a little confusing. The authors put the range in [] brackets and then abbreviated standard deviation with () brackets. On initial reading, I incorrectly assumed that the numbers in the table in () brackets were standard deviations when, in fact, they are percentages. Perhaps the authors could consider changing the caption so that the percentage is in, say, {} brackets and make the caption say that values are given as n {%} etc.

We appreciate the reviewer for pointing this out and we recognize that certain expression in the Table 1 caption is confusing. In our revised manuscript, we used n {%} to replace n (%) and deleted the abbreviated standard deviation which has not been used.

Comment 7: In the caption for Figure 2 the authors present action potentials "at steady state". Adding the pacing frequency (or cycle length) for the steady state would be useful.

We thank the reviewer for pointing this out. We agree that showing pacing frequency is important and we have made the edit in our revision.

Comment 8: In Table 2 the VT locations are compared between the EPS and the Geno-DT model. The comparison metrics listed in the table should be better described in the table caption. It is unclear if the authors compare VT locations in the AHA segments or if the specific geometric location is used. If it is a geometric location, then I would have expected to see information on the mean error distance or similar information? If it is a comparison of AHA segments, there could be a problem if a VT location was very close to the border between segments. The predicted VT location might be very close to the measured VT location but may end up in a different segment? The authors may like to clarify the methodology and/or discuss these issues.

We thank the reviewer for this comment. We recognize the need for clarification on the comparison metrics of Table 2. In the text related to Table 2, we used the wording “anatomical location” to avoid excessive repetition of mentioning AHA segments. However, we agree that reverting it back to the “AHA segment” will reduce confusion.Regarding the point of comparing exact locations the reviewer mentioned, in clinical settings, clinicians primarily rely on AHA segments to describe the VT locations during ablation and descriptions in the EP report, rather than using exact coordinates. As such, a match between our predicted AHA segments and clinical AHA segments is a direct comparison. This alignment provides a meaningful comparison and is sufficient for assisting ablation procedures.

Comment 9: In Figure 7, activation maps are shown, and the row is labelled as Induced VTs/Geno-DT. Are the colour maps from the model or the EPS measurements? The last sentence of the caption indicates they are from the measurements, but such detailed full-wall maps seem to be from a model. The authors may like to clarify what the figure shows.

We thank the reviewer for this comment. We understand the reviewer’s concern regarding the clarity of Figure 7’s caption. While we believe that the first bold sentence in the caption adequately clarifies that the results in Figure 7 are derived from the Geno-DT model, we agree with the reviewer that it is needed to further enhance the wording clarity. In response, we have made the necessary edits to the caption in our revised manuscript.

**Comments from Reviewer 3**

We thank Reviewer 3 for giving the positive assessment. Here are the responses to the comments.

Comment 1: The small sample size is a limitation but has already been acknowledged and documented by the authors.

We thank the author for this comment, and we acknowledged the small sample size as a limitation in our manuscript.

Comment 2: Another limitation is the consideration of only two of the possible genotypes in developing the cell membrane kinetics, but again has been acknowledged by the authors.

We thank the author for this comment, and we acknowledged the consideration of only two genotypes as a limitation in our manuscript. We hope to enlarge the genotype groups in our future ARVC studies.